# Fast-exchanging spirocyclic rhodamine probes for aptamer-based super-resolution RNA imaging

Daniel Englert[1], Eva-Maria Burger[1], Franziska Grün[1], Mrigank S. Verma [2], Jens Lackner[2], Marko Lampe [3], Bastian Bühler [1], Janin Schokolowski[1], G. Ulrich Nienhaus [2,4,5,6] ✉, Andres Jäschke [1] ✉ & Murat Sunbul [1] ✉

Live-cell RNA imaging with high spatial and temporal resolution remains a major challenge. Here we report the development of RhoBAST:SpyRho, a fluorescent light-up aptamer (FLAP) system ideally suited for visualizing RNAs in live or fixed cells with various advanced fluorescence microscopy modalities. Overcoming problems associated with low cell permeability, brightness, fluorogenicity, and signal-to-background ratio of previous fluorophores, we design a novel probe, SpyRho (Spirocyclic Rhodamine), which tightly binds to the RhoBAST aptamer. High brightness and fluorogenicity is achieved by shifting the equilibrium between spirolactam and quinoid. With its high affinity and fast ligand exchange, RhoBAST:SpyRho is a superb system for both super-resolution SMLM and STED imaging. Its excellent performance in SMLM and the first reported super-resolved STED images of specifically labeled RNA in live mammalian cells represent significant advances over other FLAPs. The versatility of RhoBAST:SpyRho is further demonstrated by imaging endogenous chromosomal loci and proteins.

Fluorescence microscopy has become a key technology in the life sciences[1]. Visualization of biomolecular processes with highest spatial and temporal resolution relies largely on the ability to label biomolecules selectively with bright fluorophores. Imaging specific proteins in live cells and organisms has been facilitated greatly by genetic encoding, e.g., by using GFP-type proteins[2] and self-labeling protein tags[3]. Similar approaches for imaging RNA molecules are still limited, in marked contrast to their biological importance. In recent years, fluorescent light-up aptamers (FLAPs) have emerged as promising tools for RNA visualization in live cells[4,5]. A FLAP system consists of an RNA aptamer that can be genetically fused to a target RNA of interest and a fluorogenic small-molecule probe that lights up after specific binding to the cognate aptamer. Popular FLAP systems such as

Spinach[6], Broccoli[7] and Pepper[8] are based on dyes that are essentially non-fluorescent in solution but acquire fluorescence upon aptamer binding due to effective suppression of their intramolecular dynamics. Furthermore, FLAPs such as RhoBAST, SiRA, DNB, and Riboglow have been developed that are based on photostable, bright and intrinsically fluorescent dyes, notably rhodamines[9–16]. High brightness and low apparent photobleaching of RhoBAST- and SiRA-based aptamer:dye complexes have allowed their use as genetically encoded probes for super-resolution imaging, including single-molecule localization microscopy (SMLM) and stimulated emission depletion (STED) microscopy[12,13].

RhoBAST binds the fluorophore-quencher conjugate TMR-DN (tetramethyl rhodamine-dinitroaniline)[11] with high affinity and rapid

[1]Institute of Pharmacy and Molecular Biotechnology (IPMB), Heidelberg University, Heidelberg, Germany. [2]Institute of Applied Physics (APH), Karlsruhe Institute of Technology (KIT), Karlsruhe, Germany. [3]Advanced Light Microscopy Facility, European Molecular Biology Laboratory, Heidelberg, Germany. [4]Institute of Nanotechnology (INT), Karlsruhe Institute of Technology (KIT), Eggenstein-Leopoldshafen, Germany. [5]Institute of Biological and Chemical Systems (IBCS), Karlsruhe Institute of Technology (KIT), Eggenstein-Leopoldshafen, Germany. [6]Department of Physics, University of Illinois at Urbana–Champaign, Urbana, IL, USA. ✉e-mail: uli@uiuc.edu; jaeschke@uni-hd.de; msunbul@uni-heidelberg.de

exchange kinetics[13]. The interaction between TMR and DN is greatly suppressed in complex with RhoBAST, leading to a fluorescence enhancement (Fig. 1a, b). The incessant association and dissociation of ligands from the aptamer site results in intermittent fluorescence emission. Notably, photobleaching is not a problem due to the fast and continuous ligand exchange. Working along similar principles as DNA-PAINT[17] and Peptide-PAINT[18,19], the use of RhoBAST:TMR-DN as an imaging probe enables SMLM of RNA molecules with super-resolution, and has the additional advantage of the dye lighting up upon binding. STED microscopy is an alternative super-resolution imaging method for fast image acquisition of nanoscale structures[1]. However, it is generally sensitive to photobleaching, especially in live-cell experiments[20]. Exchangeable ligands, as employed in the aptamer-based systems, can circumvent this problem[21]. The intermittent fluorescence of aptamer:fluorophore complexes is disadvantageous for STED microscopy, as the dark periods reduce the time-averaged emission intensity and thus the signal-to-background ratio. Higher fluorophore concentrations help shorten the dark periods but raise the fluorescence background due to the unbound probe. Therefore, there is great demand for bright, fluorogenic, and photostable aptamer:fluorophore pairs displaying low background fluorescence, high affinity, and fast ligand exchange kinetics that can serve as versatile tools for super-resolution RNA imaging with a variety of fluorescence imaging modalities including STED and SMLM.

Here, we introduce a novel fluorogenic probe for the RhoBAST RNA imaging system, SpyRho (Spirocyclic Rhodamine), which exploits the open-closed equilibrium for fluorescence activation (Fig. 1c, d).

The RhoBAST:SpyRho system features high brightness and fluorescence turn-on upon complex formation, as well as high photostability and fast ligand exchange kinetics. We demonstrate an excellent performance in SMLM and report the first super-resolved STED images of specifically labeled RNA in live mammalian cells.

## Results

### Design and characterization of fluorogenic spirocyclic rhodamine probes

In the quest for a brighter FLAP system, we observed that the modest fluorescence quantum yields, $\Phi_F$, of RhoBAST-bound TMR-DN and its azetidine-containing analog JF549-DN (0.57 and 0.55, respectively) result from incomplete unquenching in the complex (Fig. 1b and Supplementary Note 1). Consequently, we decided to design quencher-free rhodamine probes that exploit a shift in the open-closed equilibrium upon binding to RhoBAST for fluorogenicity (Fig. 1c, d)[22,23].

We considered two strategies for the design of quencher-free rhodamine probes for RhoBAST (Fig. 1e), (i) fluorination of the amine substituents to increase the electrophilicity of the xanthene core, and (ii) amidation of the carboxyl group at the 3-position of the benzene moiety to enhance the nucleophilicity of the internal nucleophile. Both of these modifications were reported to increase the propensity of rhodamines to switch to the colorless, non-fluorescent spirocyclic form[24–30]. Following the first strategy, we synthesized the fluorinated azetidinyl rhodamine 5C-JF525, which indeed favors the spirolactone form more than non-fluorinated 5C-JF549 (Fig. 1e and Supplementary Note 2). However, as 5C-JF525 binds RhoBAST with about 20-fold lower

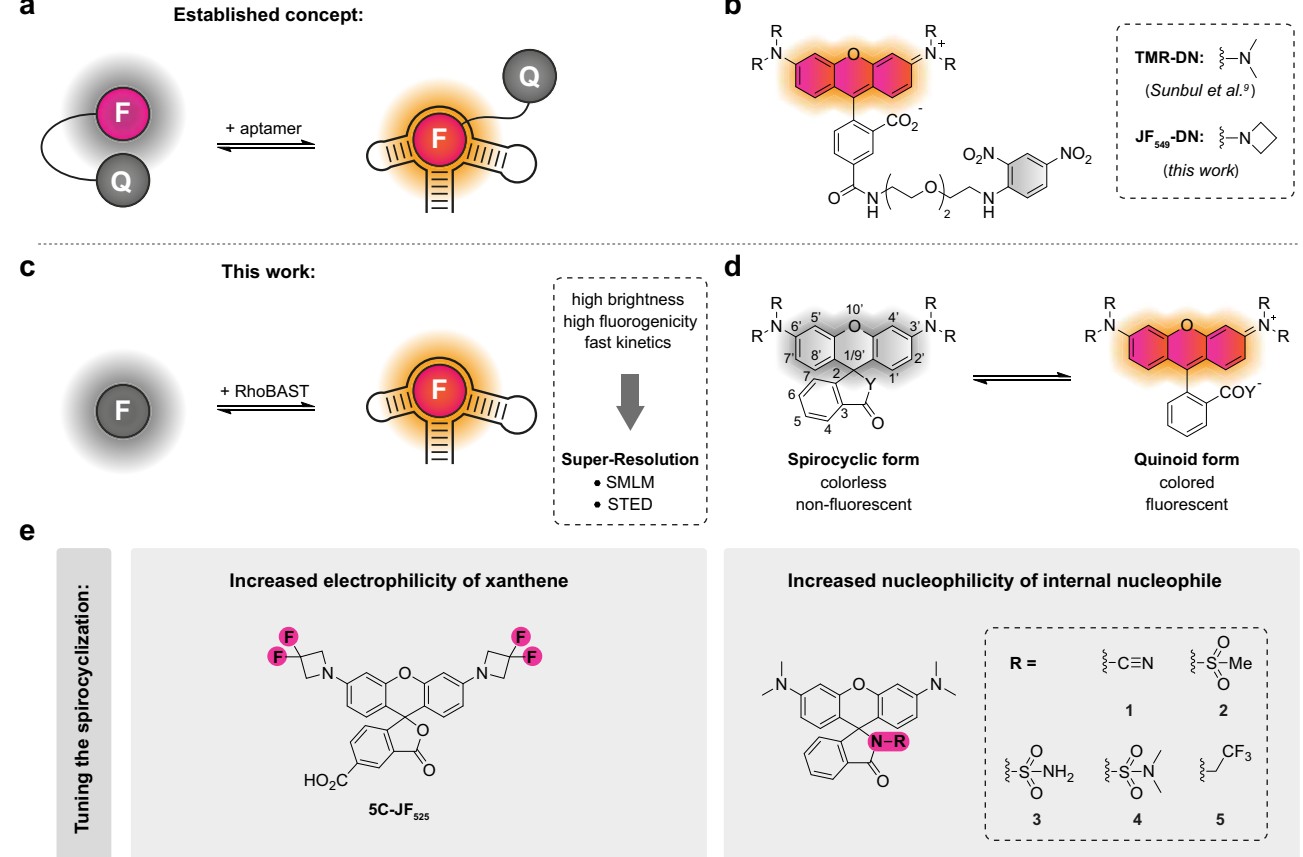

**Fig. 1 | FLAP concepts for aptamer-based RNA imaging. a** Schematic illustration of the fluorescence turn-on mechanism of fluorophore-quencher conjugates binding to their cognate aptamer. **b** Chemical structure of fluorophore-quencher conjugates used as ligands of the rhodamine-binding aptamer RhoBAST. **c** Scheme of fluorescence light-up of spirocyclic rhodamines upon binding to RhoBAST. **d** Equilibrium of rhodamines between spirocyclic (closed) and quinoid (open) form. **e** Strategies for tuning the spirocyclization equilibrium and the corresponding rhodamine derivatives synthesized in this work; left: fluorination; right: amidation.

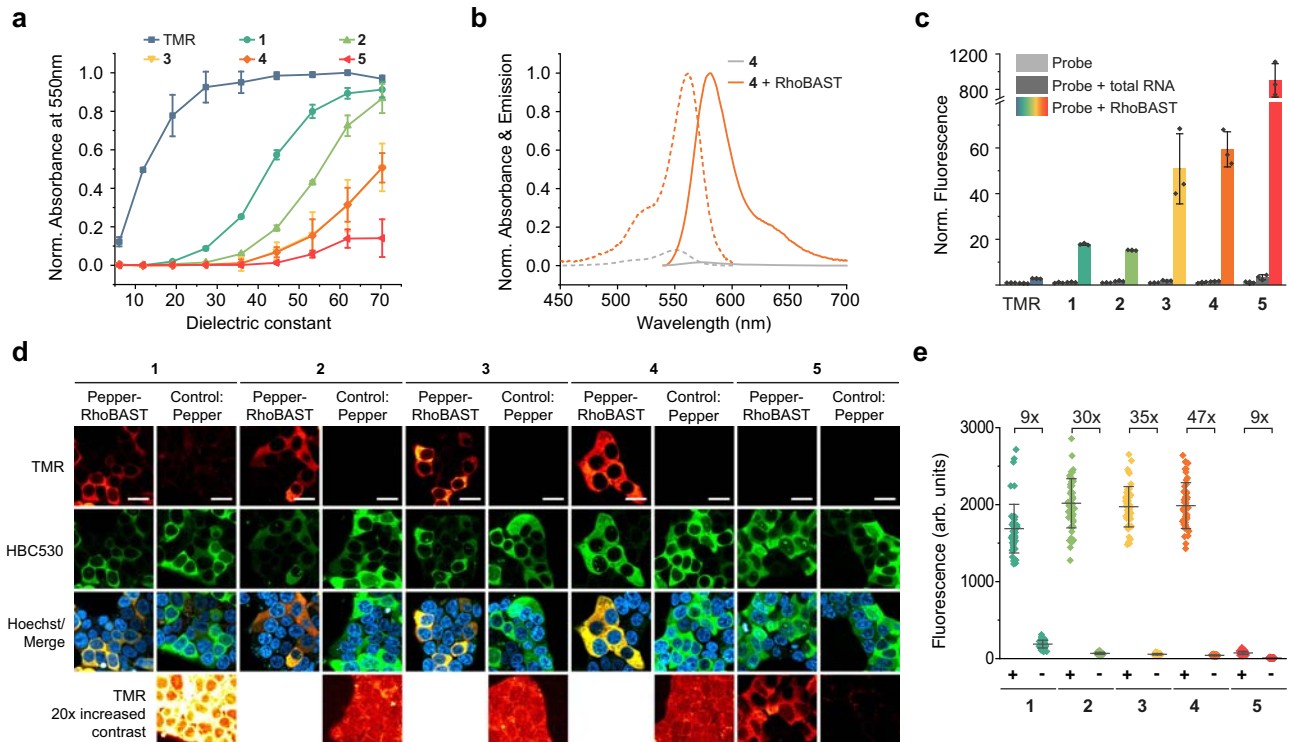

**Fig. 2 | Fluorogenic, spirocyclic rhodamines for RNA imaging. a** Absorbance of amide-functionalized rhodamine dyes and TMR (both 5 μM) as a function of the dielectric constant in solvent mixtures with different ratios of water to dioxane (10%, 20%, 30%, 40%, 50%, 60%, 70%, 80%, and 90% v/v). Data represent mean ± s.d. of three independent measurements and were normalized to the peak TMR absorbance. **b** Normalized absorption (dotted lines) and emission (solid lines) spectra of **4** (1 μM) in the presence (orange lines) and absence (gray lines) of RhoBAST (5 μM). Measurements were performed in aptamer selection buffer (ASB) at 25 °C after 1 h incubation. **c** Fluorescence turn-on values (mean ± s.d. of three independent measurements) of rhodamines (1 μM) in the presence of *E. coli* total RNA (500 ng/μL) and RhoBAST (5 μM, ca. 100 ng/μL) in ASB at 25 °C after 1 h incubation. **d** Confocal imaging of live HEK293T cells expressing circular Pepper-RhoBAST or circular Pepper aptamer incubated with rhodamines (100 nM) and HBC530 (1 μM) for 1 h. **e** Quantification of normalized fluorescence (mean ± s.d.) in the cytosol of individual transfected (Pepper-RhoBAST (+), Pepper (−)) cells (N = 50) of TMR-channel images such as those shown in **d**. Scale bars, 20 μm.

affinity than 5C-JF₅₄₉ and displays a fluorescence decrease instead of the desired light-up, this strategy was not pursued further (Supplementary Note 2).

Accordingly, we focused on the second strategy and synthesized a set of rhodamine derivatives with electron-deficient amide functionalities at the 3-position, similar to the previously reported MaP dyes (Fig. 1e)[26]. First, the absorbance values of the rhodamine dyes were measured as a function of the dielectric constant of the solvent to determine the $D_{50}$ value (dielectric constants at half-maximum absorbance), a characteristic parameter reporting on the positions of the open-closed equilibrium (Fig. 1d)[31–33]. Compared to the parent dye TMR ($D_{50} = 12$; $\varepsilon = 75{,}000\,M^{-1}cm^{-1}$), all amide-functionalized probes **1-5** displayed greater $D_{50}$ values (>40) and smaller extinction coefficients (<200 to 24,000 $M^{-1}cm^{-1}$) in aqueous solution (Fig. 2a and Supplementary Table 1), reflecting their strong tendency to form the colorless spirolactam. Encouraged by these results, we investigated if binding to RhoBAST leads to the desired shifts of the equilibrium to the fluorescent quinoid form. Indeed, upon RhoBAST addition (5 μM) to probes **1-5** (1 μM), we observed a significant absorbance (from 2.6- to >100-fold) and fluorescence (from 15- to >900-fold) increase, whereas the absorbance and fluorescence of TMR only increased by factors of 1.1 and 2.9, respectively (Fig. 2b and Supplementary Fig. 1a, b). At 5 μM RhoBAST, probes **1-4** exhibited similar absorbances (extinction coefficients, $\varepsilon = 60{,}000$–$69{,}000\,M^{-1}cm^{-1}$), while **5** showed a substantially lower absorbance due to incomplete complexation, as is confirmed by measurements of the equilibrium dissociation coefficients, $K_D$ (see below). All probes **1-5** displayed the characteristic bathochromic shifts of the absorption and emission maxima after binding to RhoBAST[13]. As with TMR ($\Phi_{F,\ bound} = 0.92$), a roughly twofold quantum yield increase

($\Phi_{F,\ bound} \geq 0.95$; $\Phi_{F,\ unbound} \leq 0.47$) was observed for **1-5** in complex with RhoBAST, contributing to the observed fluorescence enhancement. The very high quantum yields resulting in high molecular brightnesses in combination with the observed fluorogenicity are the key advantages of this quencher-free design strategy over the previously established fluorophore-quencher conjugates. The observed fluorogenicity roughly follows the trend expected from the $D_{50}$ values; a higher $D_{50}$ correlates with a greater fluorescence enhancement. Furthermore, incubation of probes **1-5** with total RNA did not result in a major fluorescence increase, demonstrating the specificity of the interaction between the probes and RhoBAST (Fig. 2c). Spectroscopic parameters of all free and RhoBAST-bound rhodamine probes are compiled in Supplementary Table 1.

RhoBAST exhibits high binding affinities towards the novel rhodamine probes **1-4**, with $K_D$ values ranging from 27 to 43 nM, except for **5** ($K_D > 10\,μM$; Supplementary Fig. 1c and Supplementary Table 1). As for the RhoBAST:TMR-DN complex, we observed rather large association ($k_{on}$ values ranging from $2.1 \times 10^7$ to $6.8 \times 10^7\,M^{-1}s^{-1}$) and dissociation ($k_{off}$ values ranging from 1.6 $s^{-1}$ to 3.6 $s^{-1}$) rate coefficients for probes **1-4** (Supplementary Fig. 2 and Supplementary Table 2).

**Performance of the new probes in no-wash live-cell RNA imaging**

For the assessment of probes **1-5** in live-cell confocal laser scanning microscopy (CLSM), we generated a circular Pepper-RhoBAST fusion aptamer (Pepper-RhoBAST), which we expressed in HEK293T cells from the U6 promoter using the Tornado expression system[34] (Supplementary Fig. 3a–c). The fluorescence of the Pepper:HBC530 system, which was detected in a separate green channel, was used to normalize the expression levels of RNA in different cells. Cells were incubated

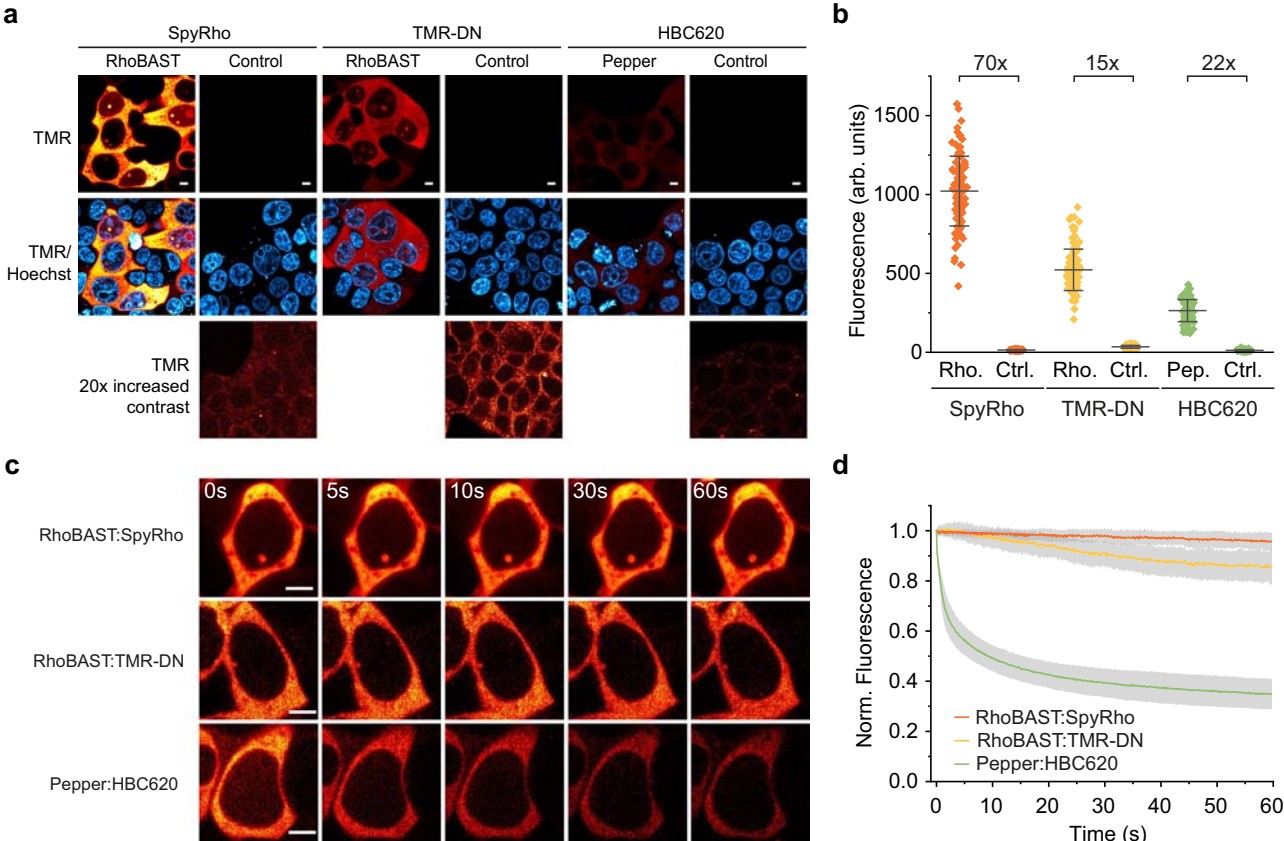

**Fig. 3 | Comparison of RhoBAST:SpyRho with other FLAPs. a** Confocal imaging of live HEK293T cells expressing circular RhoBAST or circular Pepper aptamers, incubated with the corresponding fluorogenic probes (100 nM) for 1 h prior to imaging. As control, cells expressing the aptamer (Pepper for SpyRho and TMR-DN, RhoBAST for HBC620) were used. **b** Quantification of fluorescence (mean ± s.d.) in the cytosol of individual transfected cells (*N* = 100) from TMR-channel images as those shown in **a**. **c** Imaging of live HEK293T cells expressing circular RhoBAST or circular Pepper, incubated with the corresponding fluorogenic probes (100 nM) for 1 h prior to imaging. Cells were imaged continuously at a confocal spinning disk microscope (9.7 frames/s, 1.65 mW) for 60 s. **d** Fluorescence intensity decay under continuous illumination, plotted against time and normalized to the initial value (mean ± s.d.). Data are based on the cytosolic fluorescence from individual cells (*N* = 30) as those shown in **c**. Scale bars, 5 μm.

with 100 nM of tetramethyl rhodamine probes and imaged. All probes except **5** showed a high cytosolic fluorescence in cells expressing the RhoBAST construct (Fig. 2d). Cells expressing the Pepper aptamer only were used as control. Consistent with the previous characterization, the background fluorescence in control cells decreased with increasing $D_{50}$ values, and the highest signal-to-background ratio (47-fold) was observed for the sulfamide compound **4** (Fig. 2e and Supplementary Fig. 3d). Furthermore, **4** displayed the most favorable in vitro properties such as a high fluorescence turn-on (60-fold) and brightness when complexed with RhoBAST ($\varepsilon = 65,000\ M^{-1}cm^{-1}$; $\Phi_F = 0.95$), a high binding affinity towards RhoBAST ($K_D = 34\ nM$) and fast exchange kinetics ($k_{on} = 2.1 \times 10^7\ M^{-1}s^{-1}$; $k_{off} = 1.8\ s^{-1}$). Accordingly, we focused on **4**, dubbed SpyRho (Spirocyclic Rhodamine), for further characterizations and applications.

SpyRho displays fast cellular uptake thanks to its highly membrane permeable, predominantly non-polar spirocyclic form (Supplementary Fig. 4). In fact, the appearance of fluorescence of RhoBAST in HEK293T cells was greatly accelerated upon SpyRho exposure with respect to the zwitterionic TMR-DN ligand under otherwise identical conditions (Supplementary Fig. 4c). We also examined the influence of pH on the fluorescence of SpyRho and its complex with RhoBAST, as the equilibrium between the spirocyclic and quinoid forms of rhodamines depends on pH[30,35]. We found that the fluorescence intensity of RhoBAST:SpyRho is independent of pH between 6 and 8 and only slightly decreased (by ~10%) at pH 5, and the fluorescence intensity of unbound SpyRho varies minimally between pH 5 and 8

(Supplementary Fig. 5). Therefore, the RhoBAST:SpyRho system can be reliably used to visualize RNAs in different subcellular locations within the pH range 5–8. Furthermore, we compared the brightness, signal-to-background ratio and effective photostability of RhoBAST:SpyRho with RhoBAST:TMR-DN and Pepper:HBC620, which has been reported to be the brightest and most photostable Pepper:ligand complex[8]. In live cells expressing the RhoBAST or Pepper circular aptamers, the RhoBAST:SpyRho system displayed a two- and four-fold brighter fluorescence, and a five- and threefold higher signal-to-background ratio than RhoBAST:TMR-DN and Pepper:HBC620, respectively, at 100 nM probe concentration (Fig. 3a, b). Even with a tenfold higher HBC620 concentration, RhoBAST:SpyRho was still twice as bright as Pepper:HBC620 (Supplementary Fig. 6). Comparison of the fluorescence decay of these FLAPs in live cells under continuous illumination impressively demonstrates the superb effective photostability of the rhodamine probes (Fig. 3c, d). While the fluorescence intensity of Pepper:HBC620 dropped by 65% within 60 s of continuous illumination, the decrease was only 15% and 5% for RhoBAST:TMR-DN and RhoBAST:SpyRho, respectively. Finally, we investigated the effect of SpyRho on cell viability. Incubating cells with SpyRho (up to 1 μM) for 24 h did not show any adverse effect (Supplementary Fig. 7).

## Suppression of twisted intramolecular charge transfer through binding of SpyRho to RhoBAST

Twisted intramolecular charge transfer (TICT) is a major non-radiative decay mechanism of tetraalkyl-substituted rhodamine-based

fluorophores that relies on the rotational freedom of the amine-xanthene bond. Accordingly, its suppression through chemical modifications of the substituents at the amino groups results in significantly increased $\Phi_F$ (Supplementary Fig. 8a)[36–39], reduced temperature dependence of $\Phi_F$ and enhanced photostability of the fluorophore[36,38].

The quantum yields of the RhoBAST:dye complexes ($\Phi_{F, bound} \geq 0.92$) are markedly higher than those of the corresponding free rhodamines ($\Phi_{F, free} \leq 0.47$), which strongly suggests that the TICT pathway is efficiently suppressed upon RhoBAST binding. Comparison of SpyRho with the azetidine-substituted SpyRho analog **6** (Supplementary Fig. 8b) yields further support of this claim. Indeed, free **6** showed a higher quantum yield ($\Phi_{F, free} = 0.94$) than free SpyRho ($\Phi_{F, free} = 0.44$), suggesting successful inhibition of TICT. In contrast, the RhoBAST:SpyRho and RhoBAST:**6** complexes displayed similar quantum yields ($\Phi_{F, complex} \geq 0.95$), supporting our hypothesis. Accordingly, **6** is inferior to SpyRho as a RhoBAST ligand due to its more than twice greater $\Phi_F$ of the free probe (Supplementary Fig. 8c–e), which results in roughly twofold lower fluorescence turn-on (32-fold) and signal-to-background ratio of **6** in live cells (24-fold; Supplementary Fig. 8c–e).

We further studied the temperature dependence of the fluorescence emission of the free tetramethyl rhodamines SpyRho and TMR and the azetidine-substituted 5C-JF$_{549}$ dye (Supplementary Fig. 8f). At 37 °C, the fluorescence intensities of the free SpyRho, TMR, and 5C-JF$_{549}$ dyes were only 76%, 77%, and 95%, respectively, of their values at 25 °C. The weak temperature dependence of 5C-JF$_{549}$ indicates that the TICT pathway is intrinsically inhibited. We anticipated that the steric hindrance exerted by RhoBAST on the tetramethylamine moieties of SpyRho or TMR upon binding would substantially decrease the temperature dependence of their fluorescence emission. Indeed, the fluorescence intensities of RhoBAST:SpyRho, RhoBAST:TMR, and RhoBAST:5C-JF$_{549}$ at 37 °C were still 84%, 87%, and 89% of the values at 25 °C, respectively (Supplementary Fig. 8f).

Finally, we compared the photostabilities of RhoBAST:SpyRho, RhoBAST:**6**, and Pepper:HBC620 in an in vitro photobleaching experiment. To monitor the intrinsic photostability of the complexes, we used a 25-fold molar excess of the aptamer (500 nM) over the dye (20 nM), so that bleached probes cannot be replaced by intact ones. Both RhoBAST:SpyRho and RhoBAST:**6** complexes displayed an excellent photobleaching resistance under continuous illumination, markedly superior to that of the Pepper:HBC620 complex (Supplementary Fig. 8g, h).

Taken together, the high quantum yield, the weakly temperature-dependent fluorescence, and the remarkable photostability strongly argue in favor of TICT suppression upon tetramethyl rhodamine dye binding to RhoBAST[38].

## mRNA imaging in live cells with RhoBAST:SpyRho

Next, we aimed to assess the performance of SpyRho for imaging RhoBAST-tagged mRNAs using CLSM. To this end, we fused 16 repeats of RhoBAST into the 3′ untranslated region (UTR) of green fluorescent protein (GFP) mRNAs (Fig. 4a). Live HEK293T cells expressing *GFP-RhoBAST$_{16}$* mRNA were incubated with SpyRho (100 nM) and imaged. The fluorescence in the TMR channel showed the expected, mainly cytosolic localization (Fig. 4b, c)[8,13]. Cells expressing *GFP* mRNA were used as negative control and displayed minimal background fluorescence in the TMR channel. Moreover, we systematically analyzed images of live HEK293T cells expressing *mAzurite-RhoBAST$_n$* mRNA with different numbers of RhoBAST repeats ($n = 2, 4, 8$, and 16) and obtained signal-to-background ratios of 4-, 7-, 11-, and 28-fold, respectively, in the presence of 100-nM SpyRho (Supplementary Fig. 9). Thus, mRNA can be visualized with sufficient signal-to-background ratio using only two repeats of RhoBAST. Additionally, we visualized the CGG trinucleotide repeat-containing *FMR1* (fragile-X mental retardation) mRNA. This mRNA is associated with the common genetic disease FXTAS (Fragile X-associated tremor/ataxia syndrome) that arises from an increased number of CGG repeats (55–200 copies) in the 5′-UTR of the *fmr1* gene[40,41]. The repetitive extension causes a greatly increased expression level of the *fmr1* gene[42–44], and the resulting mRNA transcripts are known to form intranuclear aggregates[13,40,41,45,46]. The toxicity of these mRNA transcripts arises, among other factors, from their excessive binding to various RNA binding proteins such as the splicing factor Sam68, which leads to sequestration and (partial) loss of function of these proteins[40,45]. To visualize the mRNA aggregates, transcripts coding for an FMR1-GFP fusion containing 99 CGG repeats in the 5′-UTR and 16 copies of RhoBAST in the 3′-UTR were expressed in Cos7 cells (Fig. 4d)[13], which displayed the characteristic nuclear aggregation pattern of this mRNA (Fig. 4e). In accordance with previous studies, co-expression of *CGG$_{99}$-FMR1-GFP-RhoBAST$_{16}$* mRNA and HaloTag7-Sam68 fusion protein in cells showed colocalization of CGG repeat-containing *FMR1* mRNA aggregates and Sam68 protein (Fig. 4e, f)[45,46]. Control cells expressing untagged CGG-containing mRNAs displayed a similar Sam68 localization pattern, whereas a more homogenous distribution of Sam68 was detected in the nuclei of cells expressing only Sam68.

## Applications beyond RNA imaging

In addition to RNA visualization, we further examined the RhoBAST:SpyRho system for live-cell imaging of endogenous chromosomal loci. To this end, the RNA-guided, nuclease-deficient Cas9 protein (dCas9) was used in combination with RhoBAST-modified single-guide RNAs (sgRNAs; Fig. 5a)[47]. Based on the design principle from previous studies, RhoBAST was inserted into the tetraloop and/or the stem loop 2 of the sgRNA (Supplementary Fig. 10a)[8,47,48]. To visualize centromeres, cells were co-transfected with a plasmid coding for a dCas9-GFP fusion protein and a second one for the centromere-targeting sgRNA. For all sgRNA constructs, numerous nuclear foci were detected in the GFP channel, indicating that RhoBAST insertion did not abolish the function of the sgRNA (Fig. 5b and Supplementary Fig. 10b). Moreover, fluorescence in the TMR channel, which colocalized with the GFP signals, was only visible in cells transfected with RhoBAST-modified sgRNAs, demonstrating the specificity of the aptamer-based imaging approach. The highest average fluorescence intensity of foci was obtained for cells expressing the sgRNA containing two copies of RhoBAST (Supplementary Fig. 10c).

To further illustrate the versatility of our RhoBAST:SpyRho system for aptamer-based imaging, we visualized proteins by exploiting the MS2 system (Fig. 5c)[49]. The protein of interest (POI) was genetically fused to a GFP-tagged synonymous tandem dimer of the RNA-binding protein MCP (stdMCP-stdGFP)[50], and the corresponding MS2 binding motif was inserted into the tetraloop of RhoBAST (Fig. 5d and Supplementary Fig. 11a, b). Several POI-stdMCP-stdGFP fusions, which localize in different subcellular compartments, specifically, the outer mitochondrial membrane (TOMM20), the endoplasmic reticulum membrane (Sec61β), and the nucleus (H2B), were co-expressed with circular RhoBAST-MS2 RNA constructs in Cos7 and U2OS cells (Fig. 5e and Supplementary Fig. 11c). Colocalization of the GFP and the TMR emission clearly demonstrates the successful tethering of the investigated proteins to the reporter RNA (Fig. 5f and Supplementary Fig. 11d).

## Fluorophore localization-based super-resolution RNA imaging

SMLM is a microscopy technique that achieves nanoscale image resolution by precise localization of individual fluorescence emitters[51]. Detection of photons from a particular fluorophore site without interfering contributions from neighboring sites places special demands on the probes used for SMLM. Oftentimes, fluorophores or fluorescent proteins are used that can switch, either spontaneously or by photoactivation, between fluorescent and non-fluorescent states. Alternatively, it is possible to decorate the sample with specific sites for

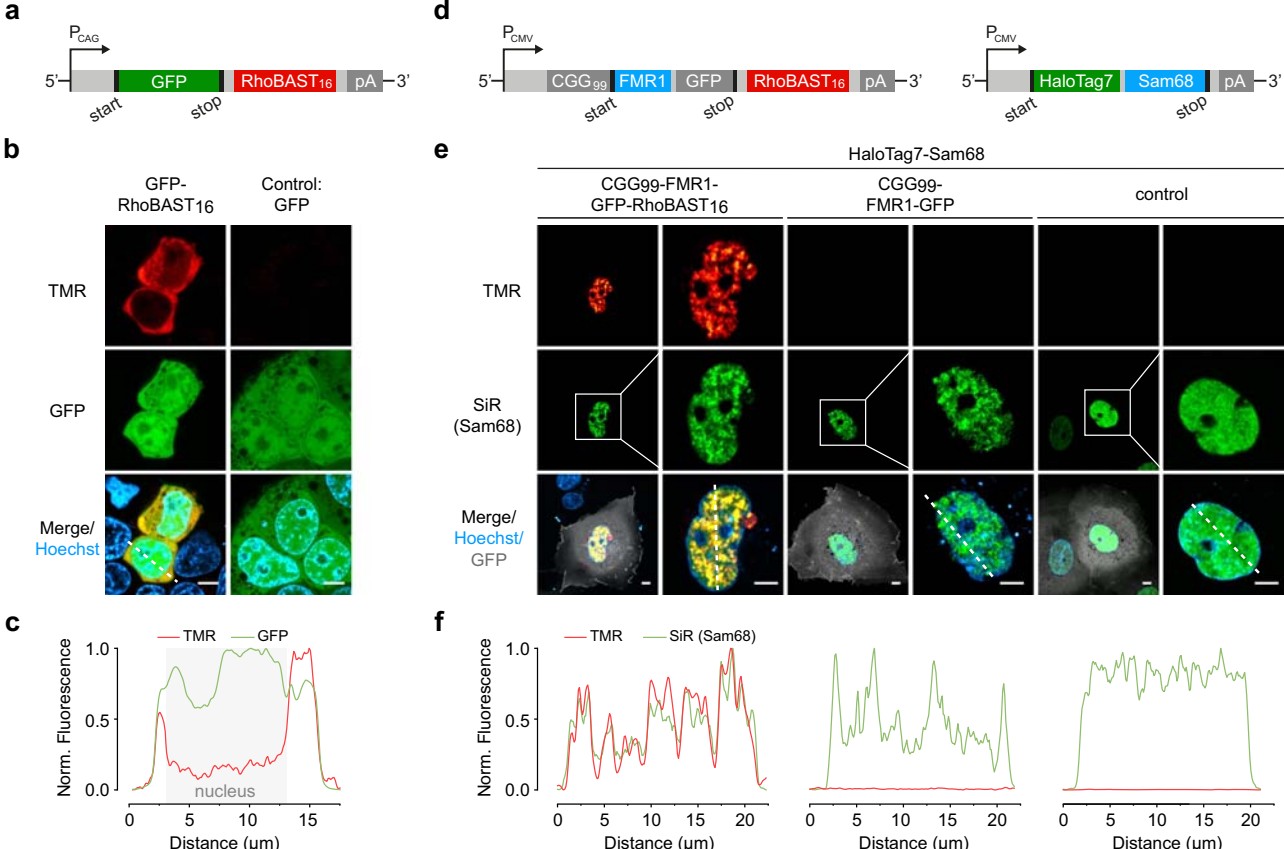

**Fig. 4 | CLSM imaging of mRNAs using RhoBAST:SpyRho. a** Schematic illustration of the construct used for expressing RhoBAST-tagged GFP mRNA. **b** Confocal images of HEK293T cells expressing GFP-RhoBAST$_{16}$ or GFP (control) mRNA incubated with SpyRho (100 nM) for 1 h before imaging. At least three independent experiments were carried out with similar results. **c** Normalized fluorescence profile along the dashed line in panel **b** (lower left). **d** Schematic illustration of the constructs used for expressing CGG-containing FMR1-GFP mRNAs fused to 16 repeats of RhoBAST and HaloTag7-Sam68. **e** Confocal images of live Cos7 cells expressing CGG$_{99}$-FMR1-GFP-RhoBAST$_{16}$, CGG$_{99}$-FMR1-GFP or GFP (control) mRNA and HaloTag7-Sam68 fusion protein (plasmid ratio 10:1) incubated with SpyRho (100 nM) and MaP700-Halo (200 nM, SiR channel) for 1 h before imaging. At least three independent experiments were carried out with similar results. **f** Normalized fluorescence profiles along the dashed lines shown in **e**. The fluorescence in the TMR channel was normalized to the highest value detected for CGG$_{99}$-FMR1-GFP mRNA. Scale bars, 5 μm.

transient fluorophore binding, so that intermittent emission from these sites is visible in the presence of background created by many freely diffusing fluorophores. By using FLAPs, we can combine these two conceptually different modes of SMLM operation. There are two key advantages of this strategy. (i) Continuous ligand exchange ensures that the number of photons collected from a particular site is not limited by photobleaching; it thus offers ultimate localization precision and best-possible image resolution. (ii) Problems with background from freely diffusing dyes are greatly reduced due to fluorescence turn-on upon binding.

The RhoBAST:SpyRho system is excellent for SMLM because it combines high affinity (selectivity) with continuous and rapid ligand exchange, which is essential for acquisition of camera frames in fast succession[52]. SpyRho binds to RhoBAST forming the fluorescence-ON state, which has a lifetime of about 0.5 s on average ($\tau_{on} = k_{off}^{-1}$). Subsequently, the ligand dissociates before another SpyRho molecule binds. This ligand-free period, the fluorescence-OFF state, lasts for about 1 s on average in the presence of 50 nM SpyRho ($\tau_{off} = (k_{on} \times c)^{-1}$, with dye concentration, $c$). The OFF state can be tuned via the SpyRho concentration. Moreover, it can be effectively shortened by tagging the RNA of interest with multiple repeats of RhoBAST. In addition to these kinetic properties, the large fluorescence turn-on of SpyRho ensures a low background arising from the unbound dye.

We exploited the superb properties of the RhoBAST:SpyRho system to visualize structural details of the nuclear trinucleotide repeat-

containing *CGG$_{99}$-FMR1-GFP-RhoBAST$_{16}$* mRNA aggregates in Cos7 cells (Fig. 6a, b). In comparison to the widefield epifluorescence image, the reconstructed SMLM image shows greatly enhanced spatial resolution (Fig. 6b, c and Supplementary Movie 1) with about 25-nm localization precision (Supplementary Fig. 12). Additionally, we imaged the phase-separated, spherical subnuclear structures formed upon over-expression of circular RhoBAST in HeLa cells by SMLM (Supplementary Fig. 13)[13].

## STED microscopy applications

As a raster-scanning microscopy method, STED microscopy is similar to CSLM. To achieve super-resolution, a donut-shaped depletion beam is applied following the regular focused excitation beam[51]. The STED method is rather sensitive to fluorophore photobleaching due to the required high power of the depletion beam[21]. The RhoBAST:-SpyRho system offers unique advantages for STED imaging because SpyRho bound to RhoBAST is intrinsically photostable, and the non-covalent nature of the system allows for constant and rapid dye exchange. Consequently, this fluorescent probe system is minimally affected by photobleaching, unlike labeling approaches using fluor-escent proteins or covalent protein labeling tags such as SNAP or Halo[20]. For STED imaging, the high fluorescence turn-on of RhoBAST:SpyRho is invaluable, allowing higher SpyRho concentra-tions without compromising the image quality by background enhancement.

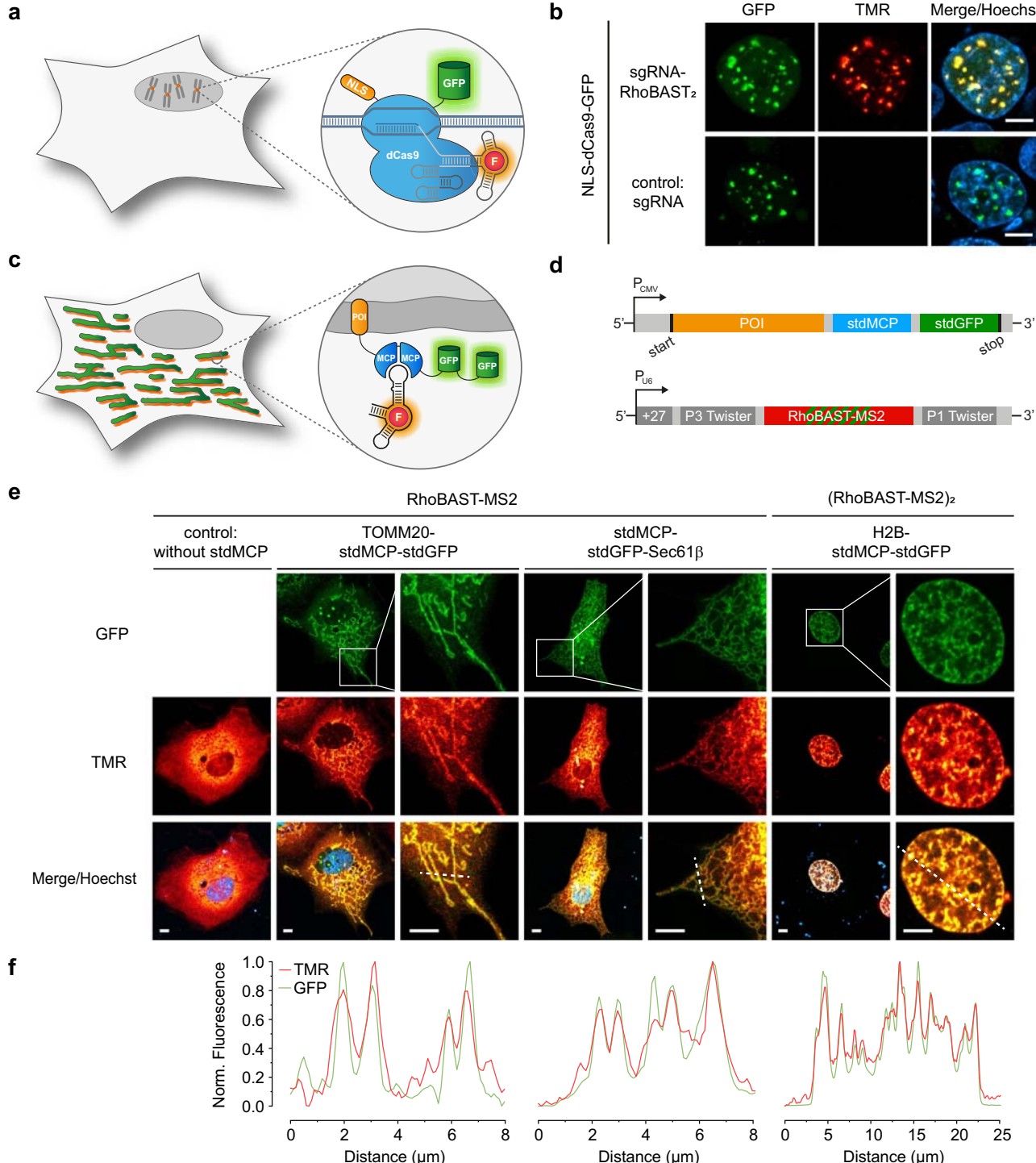

**Fig. 5 | Visualization of genomic loci and proteins using RhoBAST:SpyRho.**
**a** Illustration of the general principle of visualizing genomic loci using nuclease-deficient Cas9 protein (dCas9) and RhoBAST-modified sgRNAs. **b** Confocal imaging of HEK293T cells co-expressing NLS-dCas9-GFP and an unmodified sgRNA or sgRNA-RhoBAST$_2$ construct (plasmid ratio 1:10) targeting centromeres (targeting sequence: GAATCTGCAAGTGGATATT). Cells were incubated with SpyRho (100 nM) for 1 h prior to imaging. Maximum intensity projections of the acquired z-stacks (500 nm step size, 5 μm total) are shown for the GFP and TMR channels. At least three independent experiments were carried out with similar results. **c** Illustration of the general principle of visualizing proteins using the dimeric MS2 coat protein (MCP) and RhoBAST-MS2 RNAs. **d** Constructs used to express the protein of interest (POI) tagged with synonymous tandem dimers of MCP (stdMCP) and GFP (stdGFP), and circular RhoBAST-MS2 reporter RNA. **e** Confocal imaging of live Cos7 cells co-expressing circular RhoBAST-MS2 (for H2B-stdMCP-stdGFP: circular [RhoBAST-MS2]$_2$) and different stdMCP fusion proteins (plasmid ratio 10:1). Cells were incubated with SpyRho (100 nM) for 1 h prior to imaging. For clarity, the merged zoom-ins are shown without Hoechst. At least three independent experiments were carried out with similar results. **f** Normalized intensity profiles along the indicated dashed lines in **e**. Scale bars, 5 μm.

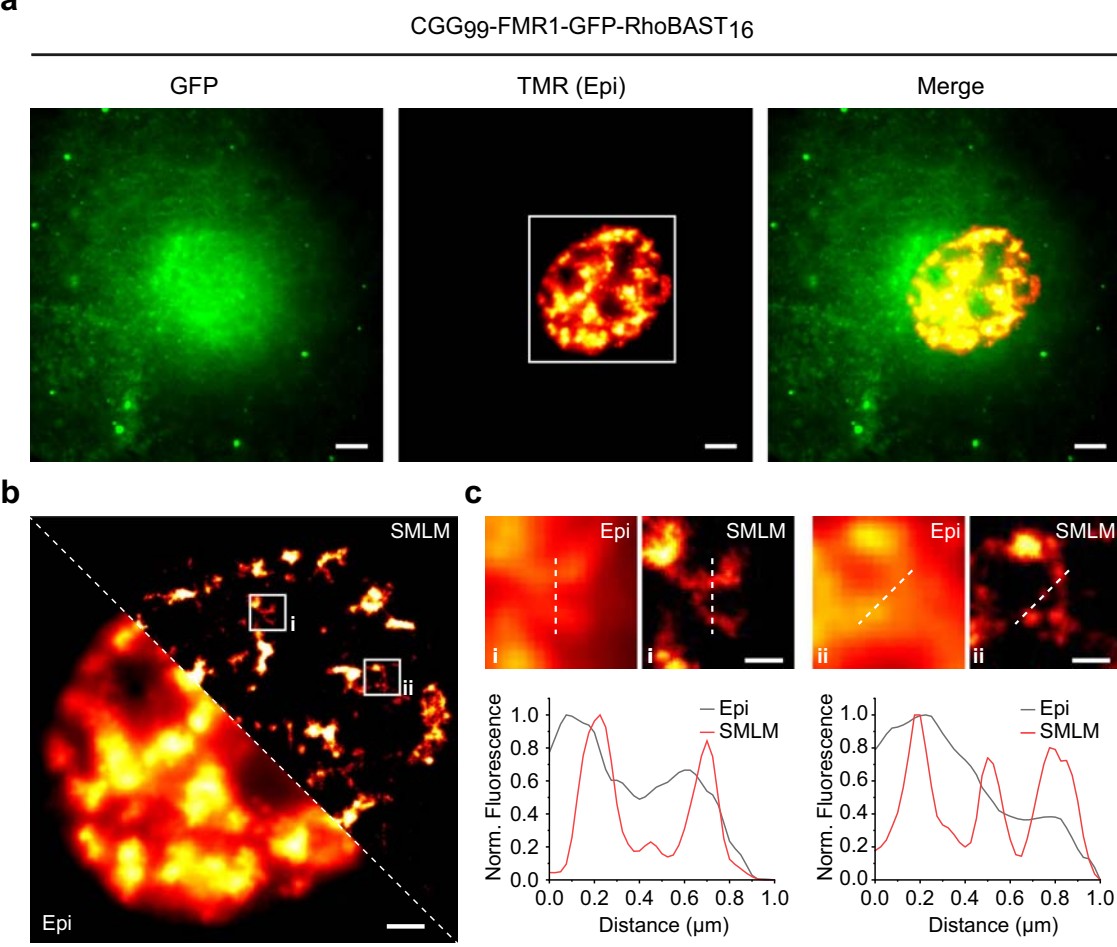

**Fig. 6 | SMLM of CGG-containing mRNA aggregates. a** Epifluorescence (Epi) images of fixed Cos7 cells expressing CGG$_{99}$-FMR1-GFP-RhoBAST$_{16}$ mRNA incubated with SpyRho (1 nM) for 30 min before imaging. At least three independent experiments were carried out with similar results. **b** Zoom-in of the epifluorescence image (TMR channel, white square) in **a** and the corresponding SMLM image, reconstructed from 80,000 frames, each with 30 ms exposure. **c** Top: close-ups of the regions indicated by the white frames in **b**; bottom: normalized intensity profiles along the white lines. Scale bars, 5 μm (**a**), 2 μm (**b**), and 500 nm (**c**).

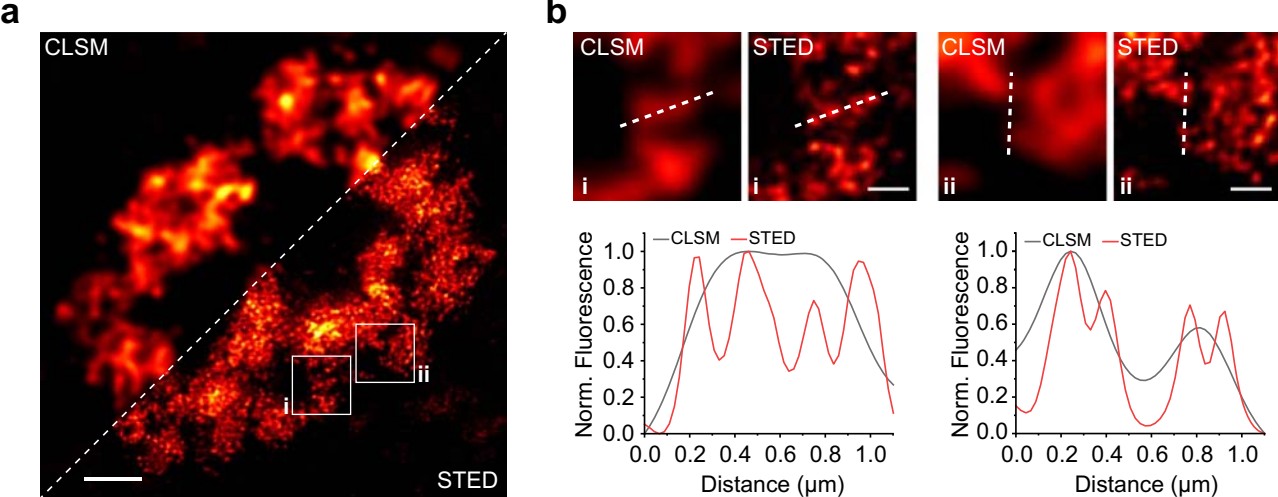

**Fig. 7 | STED microscopy of CGG-containing mRNA aggregates in live cells.**
**a** Combined CLSM and STED image (deconvolved) of a nucleus of a live Cos7 cell expressing CGG$_{99}$-FMR1-GFP-RhoBAST$_{16}$ mRNA, which was incubated with SpyRho (100 nM) for 1 h prior to imaging. At least three independent experiments were carried out with similar results. **b** Top: close-ups of regions marked by the white squares in **a**); bottom: normalized intensity profiles along the dashed lines depicted in the images. Scale bars, 2 μm (**a**) and 500 nm (**b**).

To demonstrate the beneficial properties, photobleaching in live-cells was compared for RhoBAST:SpyRho and proteins covalently labeled with MaP555 derivatives[26], which are based on the identical fluorophore core as SpyRho (Supplementary Fig. 14a). For this comparison, circular RhoBAST (cytosolic localization) and a HaloTag-SNAP-Tag protein fused to a nuclear localization sequence were co-expressed in HEK293T cells and labeled with the corresponding rhodamine derivative (Supplementary Fig. 14b, c). Repeated imaging of these cells clearly shows the significantly higher bleaching resistance of the cytosolic RhoBAST:SpyRho complex compared to covalently MaP555-labeled protein tags localized in the nucleus (Supplementary Fig. 14d). This experiment strikingly illustrates the advantage of non-covalent labeling approaches in combination with the enhanced photostability of the aptamer-bound dye over conventional labeling techniques.

Finally, we used RhoBAST:SpyRho to collect super-resolved images of trinucleotide repeat-containing $CGG_{99}$-$FMR1$-$GFP$-$RhoBAST_{16}$ mRNA aggregates in live Cos7 cells using STED microscopy with two-dimensional STED confinement (2D-STED, Fig. 7 and Supplementary Fig. 15a) as well as 3D-STED (Supplementary Fig. 15d, e). To the best of our knowledge, live-cell STED microscopy of RNAs was previously only shown in $E.$ $coli$ using the SiRA FLAP system[12], so here we report the first STED images of specifically labeled RNAs in live mammalian cells. Thanks to the high photostability of the RhoBAST:SpyRho system, we could record multiple frames of the same region of interest with negligible loss of fluorescence intensity (Supplementary Fig. 15b, c). Notably, STED images obtained with SpyRho were substantially brighter than those obtained with TMR-DN (Supplementary Fig. 15f) highlighting the advantage of SpyRho over TMR-DN.

## Discussion

We have described the development of fluorogenic, bright, photostable and fast-exchanging rhodamine dyes as novel ligands for the rhodamine-binding aptamer RhoBAST. These probes rely on the environment-dependent equilibrium of rhodamines between a non-fluorescent spirocyclic form and a fluorescent quinoid form. Thanks to their enhanced brightness, fluorescence turn-on and effective photostability, these probes constitute a substantial advance over previous ligands and allow for facile super-resolution RNA imaging with SMLM and STED microscopy. We found strong evidence that suppression of the non-radiative decay process TICT upon binding of SpyRho to RhoBAST is responsible for the excellent fluorescence quantum yield and high photostability of the RhoBAST:SpyRho complex.

The remarkable in vitro photophysical properties of the Rho-BAST:SpyRho system were maintained in live-cell experiments, with stronger fluorescence and greatly enhanced signal-to-background ratio in comparison to its quencher-based predecessor RhoBAST:TMR-DN. Its high photostability and the continuous dye exchange render RhoBAST:SpyRho resistant to photobleaching. It is a valuable tool for imaging RhoBAST-tagged mRNAs, and can further be used to visualize endogenous chromosomal loci and proteins. Most exciting, however, has been the performance of the RhoBAST:SpyRho system in super-resolved RNA imaging using two different super-resolution modalities, SMLM and STED. SMLM was facilitated by the fast dye exchange kinetics resulting in intermittent fluorescence emission, together with the enhanced brightness and fluorescence turn-on of SpyRho. Apart from these factors, the extraordinary bleaching resistance of the RhoBAST:SpyRho system allowed us to acquire super-resolved STED images of specifically labeled RNA molecules in live mammalian cells for the first time. Clearly, RhoBAST:SpyRho constitutes a significant advance for in live-cell imaging over previously developed FLAPs including RhoBAST:TMR-DN and Pepper:HBC620. We are confident that the excellent features of this novel imaging tool will inspire many new applications of RhoBAST:SpyRho in advanced fluorescence microscopy.

## Methods

### General

Reagents were purchased from Thermo Fischer, New England Biolabs or Sigma-Aldrich and used without further purification. All applied fluorescent dyes based on rhodamine were synthesized in house (see Supplementary Note 3). SpyRho can be obtained from Spirochrome. Stock solutions of dyes in DMSO were stored at $-20\,°C$ and the final DMSO concentration in all experiments was $\leq1\%$. All DNA oligonucleotides were purchased from Integrated DNA Technologies. Oligonucleotide concentrations were determined using a NanoDrop One UV/Vis spectrometer (Thermo Fischer). Deionized water used for all experiments was purified with a Milli-Q Synthesis A10 Water Purification system (Merck). Absorbance spectra were recorded on a Cary50 UV/Vis spectrometer (Varian) at ambient temperature. Fluorescence measurements were performed on a JASCO FP-650 or FP-850 fluorometer equipped with a temperature controller set to $25\,°C$.

### Cell culture

Eukaryotic HEK293T (ACC 635), U2OS (ACC 785), HeLa (ACC 57), and Cos7 (ACC 60) cells were obtained from DSMZ and cultured at $37\,°C$ and $5\%$ $CO_2$ in Dulbecco's Modified Eagle's Medium (DMEM; without phenol red, high glucose, HEPES and glutamine; Thermo Fischer) supplemented with FBS (10%), penicillin (100 U/ml) and streptomycin (100 μg/ml). Cells were split every 2–3 days or at confluency.

### Chemical synthesis

A detailed description of the chemical synthesis and characterization of novel dyes can be found in Supplementary Note 3.

### Extinction coefficient and fluorescence quantum yield

Concentrations of dye stock solutions were determined from three independent $^1H$ NMR measurements using an internal standard with a known concentration. To obtain the peak extinction coefficient, $\varepsilon$, absorbance spectra were measured in Aptamer Selection Buffer (ASB; 20 mM HEPES (pH 7.4), 1 mM $MgCl_2$, 125 mM KCl) at three or more concentrations; and $\varepsilon$ was calculated from the peak absorbance using the Lambert–Beer law.

Absolute fluorescence quantum yields were measured on a Quantaurus QY C11347 spectrometer (Hamamatsu). All measurements were performed with dilute dye solutions (absorbance $\leq 0.1$) in ASB at ambient temperature.

### $D_{50}$ value

Absorbance spectra of dyes (5 μM) were recorded in different water-dioxane mixtures (10%, 20%, 30%, 40%, 50%, 60%, 70%, 80%, and 90%; v/v; three independent measurements for each mixture). The absorbance was normalized to the measured maximal absorbance of the individual derivative or TMR (for **1-5**) and plotted against the dielectric constant of the water-dioxane mixtures. $D_{50}$ values were obtained by determining the dielectric constant at half-maximum absorbance.

### pH dependency of absorbance

Absorbance spectra of dyes (5 μM) were recorded in 0.1 M phosphate buffer (prepared using $H_3PO_4$, $NaH_2PO_4$, $Na_2HPO_4$, $Na_3PO_4$ in different ratios and NaOH to adjust the pH) over wide pH ranges (up to 2–12); three independent measurements were made for each pH. The peak absorbances were normalized to the peak TMR value and plotted against pH.

### In vitro transcription of RhoBAST

The double-stranded DNA template (5'-TC<u>TAATACGACTCACTA</u>-<u>TA</u>GGAACCTCCGCGAAAGCGGTGAAGGAGAGGGCGCAAGGTTAACCGC CTCAGGTTCC-3'; T7 promotor region is underlined) for the in vitro transcription of RhoBAST was prepared by polymerase chain reaction (PCR) using Taq DNA polymerase (lab prepared stock). In vitro

transcriptions were performed using 1 μM dsDNA template, 50 ng/μL T7 RNA polymerase (lab prepared stock) and 2 mM NTPs in transcription buffer (40 mM Tris-HCl (pH 8.1), 1 mM Spermidine, 22 mM $MgCl_2$, 0.01% Triton X-100, 5% DMSO, 10 mM DDT). The transcription mixture was incubated at 37 °C for 4 h and treated with 50 U/mL DNAse I (Roche) at 37 °C for 30 min. The reaction was quenched by the addition of one volume of gel loading buffer (100 mM Tris-Borate (pH 8.3), 2 mM EDTA in 90% formamide) and purified by denaturing polyacrylamide gel electrophoresis. The appropriate RNA band was excised and eluted in 0.3 M NaOAc (pH 5.5) at 19 °C overnight. Gel pieces were removed by filtration and the RNA was precipitated with isopropanol. The obtained pellet was dissolved in deionized water and desalted using 10k Amicon Ultra centrifugal filters (Merck).

### In vitro characterization of RhoBAST:dye complexes

To ensure optimal folding, before each in vitro experiment, the RNA aptamers were treated as follows. The RNA was diluted in water, incubated at 75 °C for 2 min and cooled down to 25 °C in -10 min. Next, the appropriate buffer concentrate (6x) was added and the sample was incubated at 25 °C for an additional 10 min before use. If not stated otherwise, measurements were performed in ASB (20 mM HEPES (pH 7.4), 1 mM $MgCl_2$, 125 mM KCl).

To determine the fluorescence turn-on value, fluorescence measurements of the dyes (1 μM) were performed in the presence and absence of RhoBAST (5 μM) after incubation at room temperature for 1 h. The excitation and emission maxima of the corresponding RhoBAST:dye complex were chosen as wavelengths for excitation and detection, respectively. The turn-on value was defined as the ratio of fluorescence intensities of the aptamer:dye complex and the free dye.

Equilibrium dissociation coefficients ($K_D$ values) of RhoBAST:dye complexes were determined through measurements of the fluorescence change of the dyes upon binding to the aptamer. First, the fluorescence intensity was measured as a function of the aptamer concentration using a fixed dye concentration (10 nM, 50 nM for **5**). Fluorescence measurements were performed in ASB supplemented with 0.05% (v/v) Tween 20 at 25 °C. To determine $K_D$ values, the measured intensities were fitted to Eq. (1)[13] using OriginPro (2015):

$$F = F_0 + (F_\infty - F_0)\frac{(K_D + P_0 + [Apt]) - \sqrt{([Apt] - P_0)^2 + K_D \times (K_D + 2[Apt] + 2P_0)}}{2P_0},$$

(1)

with measured fluorescence intensity, $F$, fluorescence of the free dye, $F_O$, fluorescence upon saturation, $F_\infty$, initial dye concentration, $P_O$, and aptamer concentration $[Apt]$.

To obtain $\varepsilon$ values of RhoBAST:dye complexes and free dyes in ASB, absorbance spectra of the fluorophores (1 μM) were recorded in the presence and absence of RhoBAST (5 μM) after incubation at room temperature for 1 h.

The quantum yields, $\Phi_{F, complex}$, of RhoBAST:dye complexes were determined in relative measurements using a sulforhodamine 101 solution ($\Phi_{F, ref} = 0.95$ in ethanol)[53] as a reference. Absorbance and fluorescence emission spectra (excitation wavelength 528 nm) were measured at four or more different dye concentrations (absorbance ≤ 0.1) using a constant RhoBAST concentration (5 μM). Next, the areas under the fluorescence spectra were plotted against the absorbance (at 528 nm) and the slopes, $m$, were determined by fitting a line to the data points. The quantum yields of the complexes were determined using Eq. (2):

$$\Phi_{F,complex} = \Phi_{F,ref}\left(\frac{m_{complex}}{m_{ref}}\right) \times \left(\frac{n_{complex}^2}{n_{ref}^2}\right),$$

(2)

with $n$ denoting the refractive index of the respective solvent.

The pH dependence of the fluorescence emission intensities of the dyes (50 nM) was studied in the absence and presence of RhoBAST (1 μM) in a modified ASB (20 mM phosphate buffer, 1 mM $MgCl_2$, 125 mM KCl) at 25 °C, with pH varying between 4 and 8. The fluorescence emission intensities were normalized to the maximum fluorescence of the corresponding RhoBAST:dye complex.

In vitro photobleaching experiments of RhoBAST:dye complexes were performed as described previously[13]. Briefly, RhoBAST:dye complexes were prepared in ASB supplemented with 0.05% (v/v) Tween 20 by mixing dye and RhoBAST solutions to yield concentrations of 20 and 500 nM, respectively. After incubation at room temperature for 30 min, the whole volume of the sample (50 μL) was irradiated in an airtight sealed quartz cuvette using a focused Luxeon Rebel ES Lime LED (nominal wavelength = 567 nm, 10.2 mW; operated using Labview). Samples were irradiated for 3 h and the fluorescence intensity was measured every 20 min at 25 °C. The experiment was performed three times for each RhoBAST:dye complex.

The association and dissociation kinetics of RhoBAST:dye complexes were determined in ASB supplemented with 0.05% (v/v) Tween 20 using stopped-flow experiments as described previously[13]. Briefly, kinetic properties were determined using an SX-18 M stopped-flow spectrometer from Applied Photophysics. Each dye (5 nM) was titrated with different concentrations of RhoBAST at 25 °C and the fluorescence intensity was measured. To determine the observed kinetic rate $k_{obs}$ the time-dependent fluorescence F(t) was fitted using Eq. (3):

$$F(t) = (F_0 - F_\infty)e^{-k_{obs}t} + F_\infty$$

(3)

with $F_O$ as the initial and $F_\infty$ as the final fluorescence intensity. The association rate coefficient $k_{on}$ and the dissociation rate coefficient $k_{off}$ were extracted by plotting $k_{obs}$ versus RNA concentration and linear fitting according to Eq. (4):

$$k_{obs} = k_{on}[RNA] + k_{off}$$

(4)

### DNA cloning

Sequences of all constructs can be found in Supplementary Table 3 and sequences of all used oligonucleotides can be found in Supplementary Tables 4 and 5.

Plasmids expressing RNA constructs embedded into the Tornado system were prepared by restriction enzyme cloning. Therefore, DNA inserts containing the aptamer sequence (RhoBAST, Pepper, Pepper-RhoBAST, RhoBAST-MS2, and [RhoBAST-MS2]₂) flanked by NotI and SacII restriction sites were prepared by PCR. The inserts were double-digested and cloned into the double-digested pAV-U6 + 27-Tornado-Broccoli plasmid (Addgene, plasmid #124360).

To prepare the pcDNA5-FRT/TO-NLS-dCas9-NLS-GFP plasmid, the NLS-dCas9-NLS-GFP gene was amplified from plasmid pSLQ1658-dCas9-EGFP (Addgene, plasmid #51023) by PCR and inserted into the BamHI and XhoI restriction sites of the pcDNA5-FRT/TO plasmid (Invitrogen).

The pcDNA3-HaloTag7-Sam68 plasmid was prepared using the NEBuilder HiFi DNA Assembly kit (NEB). The vector backbone and the HaloTag7 insert were amplified by PCR using the pcDNA3-HA-Sam68-WT vector (Addgene, plasmid #17690) and pcDNA5-FRT- Halo-SNAP-NLS plasmid (kindly provided by the Johnsson group, Heidelberg) as templates.

The pAV-U6-centromere-sgRNA plasmids (sgRNA, sgRNA-RhoBAST-1, sgRNA-RhoBAST-2, sgRNA-RhoBAST₂) were prepared by inserting a double-digested gene fragment containing the U6 promotor sequence, the sgRNA construct and a polyT stretch for termination into the BamHI and XhoI restriction sites of the pAV-U6 + 27 plasmid (Addgene, plasmid #25709).

Plasmids expressing stdMCP-fusion proteins were generated using the NEBuilder HiFi DNA Assembly kit (NEB). To obtain plasmids carrying an stdGFP-tagged stdMCP gene, the vector backbone was amplified from the pcDNA5-FRT plasmid (Invitrogen), the stdMCP-stdGFP fragment was amplified from the pUbC-NLS-HA-stdMCP-stdGFP plasmid (Addgene, plasmid #98916) and the localization sequences were ordered as gene block (H2B, Sec61β) or prepared by PCR (TOMM20). The three fragments were assembled according to the manufacturer's protocol to obtain the pcDNA5-FRT-H2B-stdMCP-stdGFP, pcDNA5-FRT-TOMM20-stdMCP-stdGFP, and pcDNA5-FRT-stdMCP-stdGFP-Sec61β plasmids.

To prepare mAzurite-RhoBAST$_n$ plasmids, a stop codon was inserted into the mAzurite-C1 plasmid (Addgene, plasmid #54583) in front of the multiple cloning site. The RhoBAST repeats ($n = 2, 4, 8, 16$) were amplified from the corresponding pTac-Cat-RhoBAST$_n$ plasmid[13]. The amplified inserts were double-digested with BamHI and SalI and cloned into the double-digested modified mAzurite plasmid.

### Cell viability assay

For this study, we used the Cell Titer96 AQ$_{ueous}$ One Solution Cell Proliferation Assay (Promega) according to the manufacturer's protocol. Briefly, HEK293T cells were cultured in a clear bottom 96-well plate in supplemented DMEM (without phenol red) and incubated with SpyRho (stock solutions in DMSO, 1% final DMSO concentration) or with DMSO (control, 1% final concentration) for 24 h. Next, 20 μL of the Cell Titer96 AQ$_{ueous}$ One Solution was added to each well (100 μL) and cells were incubated for another 3 h. The absorbance at 450 nm was measured using a Tecan Safire2 plate reader, and the values of treated cells and untreated control cells were compared.

### Live-cell confocal microscopy

For imaging experiments, cells were seeded into Ibidi μ-Slides (glass bottom) with 8 or 18 wells and incubated overnight in 300 or 100 μL medium, respectively. For HEK293T cellular imaging, slides were coated with poly-D-lysine before use. On the next day, transient transfection of the appropriate plasmid(s) was performed using FuGENE HD (Promega) according to the manufacturer's procedure. After incubation overnight, the medium was exchanged with fresh DMEM. Cells were incubated in total for 48 h after transfection; subsequently, the medium was exchanged to Leibowitz (L15) medium containing the appropriate dyes. Samples were incubated for 1 h at 37 °C before imaging.

If not stated otherwise, imaging experiments were performed on a point scanning Nikon A1R confocal microscope equipped with a hybrid scanner (galvo/resonant), 4 channel detection (with GaAsP detectors), and a live-cell imaging chamber with controlled temperature and humidity. All images were acquired using a Nikon N Apo 60× NA 1.4 λs OI (WD 0.14 mm, FOV 0.21 × 0.21 mm$^2$) objective. Hoechst 33342 and mAzurite were excited at 405 nm; the emission was detected via a 450 ± 25 nm filter. GFP and HBC530 were excited at 488 nm; the emission was detected via a 525 ± 25 nm filter. TMR derivatives and HBC620 were excited at 561 nm; the emission was detected via a 595 ± 50 nm filter. Draq5 and SiR were excited at 640 nm; the emission was detected via a 700 ± 37.5 nm filter. Images were acquired using the Nikon NIS Elements software (5.11.01).

Spinning disk confocal microscopy was performed with an ERS-VoX system (PerkinElmer), based on an automated inverted Nikon Ti microscope equipped with a Hamamatsu C9100-02 EMCCD camera, a Yokagawa CSU-X1 confocal scanning unit and a temperature-controlled live-cell imaging chamber. TMR derivatives and HBC620 were excited at 561 nm; the emission was detected via a 445 ± 30 nm / 615 ± 35 nm blue/red dual-pass filter. Experiments were performed using a Nikon Plan Apo VC 100× NA 1.4 Oil objective and images were recorded at a resolution of 1024 × 1024 pixels with an exposure time of 100 ms (9.7 frames/sec) utilizing the Perkin Elmer Volocity software.

Images were processed and analyzed by the open-source application Fiji[54].

### Single-molecule localization microscopy

Single-molecule localization microscopy was performed on a home-built widefield microscope based on an Axio Observer Z1 frame (Zeiss) equipped with an Ixon Ultra X-7759 EMCCD camera (Andor) with a pixel size of 109 × 109 nm$^2$ and a quad-band dichroic mirror (z 405/473/561/640, AHF). Lasers emitting at 473 nm (Gem 473, Laser Quantum) and 561 nm (Gem 561, Laser Quantum) were used in combination with an acousto-optic tunable filter (AOTFnC-400.650, A-A Opto-Electronic) and two achromatic lenses (focal lengths 10 mm and 100 mm, Thorlabs) to expand the laser beam. Images were acquired using a Zeiss α Plan Apo 63× NA 1.46 Oil Corr M27 TIRF objective. GFP was excited with the 473 nm laser (~200 μW) and the signal was collected through a 525 ± 45 nm emission filter. TMR derivatives were excited with the 561 nm laser (~20 mW) and the signal was collected using a combination of 561 nm long-pass and 607 ± 35 nm emission filters. Typically, 80,000 frames (30 ms exposure time) were acquired and used for reconstruction of the SMLM images. SMLM analysis was performed with our a-livePALM software[52] (MATLAB 2019b); image drift was corrected as previously described[13].

### Live-cell STED microscopy

STED microscopy was performed using a Leica SP8 STED 3X microscope (Leica Microsystems, Mannheim, Germany) equipped with an 80 MHz white light laser, three depletion lasers (592, 660 and 775 nm), HyD detectors and an in-house developed temperature-controlled live-cell imaging incubator (EMBL Heidelberg, Heidelberg, Germany). SpyRho was excited at 561 nm and depleted by using the 660 nm CW-STED laser; the detection gate was open from 2 to 6 ns for 2D-STED and 1.5 to 6 ns for 3D-STED. All experiments were performed using a Leica HC PL APO 86×/1,20 W motCORR STED white water immersion objective. Images were acquired using the LAS X software version 3.5.7.23225 (Leica Microsystems, Mannheim, Germany). Images were processed and analyzed using Fiji[54]. Deconvolution was performed using Huygens Professional version 22.04 (Scientific Volume Imaging B.V., Hilversum, The Netherlands).

### Reporting summary

Further information on research design is available in the Nature Portfolio Reporting Summary linked to this article.

## Data availability

The data that support the findings of this study are available within the paper and its Supplementary Information. The full image dataset is available from the corresponding author upon request.

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

## Acknowledgements

A.J. and M.S. were supported by the Deutsche Forschungsgemeinschaft (DFG grant no. Ja794/11), A.J. and G.U.N. by the Baden-Württemberg Stiftung (MET-ID21-Aptamer-PAINT) and G.U.N. by the Helmholtz Association (Program Materials Systems Engineering) and by the DFG (GRK 2039). The authors thank K. Johnsson and L. Wang for providing the Halo-SNAP-NLS plasmid and MaP dyes, and thank the Nikon Imaging Center, Heidelberg for granting access to their facilities. The authors acknowledge Euro-BioImaging (www.eurobioimaging.eu) for providing access to imaging technologies and services via the EMBL Node (Heidelberg, Germany). We thank especially the Advanced Light Microscopy Facility (ALMF) at the European Molecular Biology Laboratory (EMBL) and Leica for their support, and gratefully acknowledge M. Mayer and L. Rohland for assistance with stopped-flow measurements. For the publication fee we acknowledge financial support by DFG within the funding programme "Open Access Publikationskosten" as well as by Heidelberg University.

## Author contributions

D.E., A.J., G.U.N., and M.S. designed the study. E.M.B. and D.E. performed the synthesis. E.M.B., D.E., and F.G. performed in vitro characterizations. D.E. prepared all plasmid constructs. D.E., F.G., B.B., and J.S. conducted confocal microscopy experiments. M.S.V. and J.L. performed and analyzed the SMLM experiments. M.L. and D.E. performed STED measurements. D.E. wrote the initial draft of the manuscript and all authors participated in revising and editing.

## Funding

## Competing interests

The authors declare no competing interests.
