## [Peer Review File · Nature Communications]

Reviewers' Comments:

Reviewer #1:

Remarks to the Author:

The manuscript by Englert et al reports the synthesis and characterization of a series of fluorogenic rhodamine dyes as ligands for the aptamer RhoBAST and their subsequent RNA visualization in live cells using different imaging modalities, including super-resolution microscopy. The authors systematically screened the dyes to select a bright, photostable, and fast-exchanging core, which switches on and off due to the equilibrium between the non-fluorescent spirocyclic form and the fluorescent quinoid form. The experiments are very well executed, and the manuscript is well written with excellent scholar presentation.

These probes constitute a remarkable improvement over the existing toolbox of RNA imaging dyes and will open several avenues for super-resolution RNA imaging. Notably, the RhoBAST:SpyRho system is compatible with live-cell experiments and potentially over long periods of time due to its high photostability. The authors provided good proof of the utility of their system by imaging endogenous chromosomal loci and proteins as well as labelled RNA molecules in live mammalian cells, which I haven't seen reported before. I would only suggest the authors a couple of minor revisions: 1) the introduction covers well the state-of-the-art for nucleic acid-based super-res imaging (eg DNA-PAINT), but it may be worth including a mention to peptide-PAINT strategies, including some recent advances in new fluorogenic building blocks (Angew Chem 2022, doi: 10.1002/anie.202216231) ; 2) I couldn't find data plots on the localization precision that is obtained in STED with this system (eg Fig 6). This would be particularly useful to know the potential application of this technology to study nanoaggregates or nanoclusters.

Overall, this is an exceptional manuscript and I recommend publication in Nature Communications. I congratulate the authors for their work. This is one of the best manuscripts I have reviewed this year.

Reviewer #2:

Remarks to the Author:

In this manuscript, Sunbul and coworkers report a novel fluorogenic probe for the RhoBAST RNA imaging system called SpyRho. This design exploits the open-closed equilibrium for fluorescence activation. Compared to RhoBAST:TMR-DN, the RhoBAST:SpyRho system features higher photostability, high brightness, higher signal-to-background ratio, and faster ligand exchange kinetics while retaining the fluorescence turn-on function upon complex formation.

RhoBAST:SpyRho system is used to visualize RNA, chromosomal loci, and proteins. Super-resolution RNA imaging with SMLM and STED microscopy in live mammalian cells is also demonstrated. The study is well performed and the conclusions drawn are well supported by the data obtained in the comprehensive experiments. Super-resolution fluorescence microscopy of cellular structures provides new insights into biology. The SpyRho probe reported in the manuscript is novel and is potentially useful in super-resolution imaging. The manuscript is thus recommended for publication in the journal after the following concerns have been addressed.

1. Most reaction conditions are not listed as containing an organic solvent. Is SpyRho water soluble at the high stock concentrations needed? If the stocks are dissolved in an organic solvent, what is its final concentration in the working solutions? What are the storage conditions such as temperature and concentration for SpyRho stocks?
2. The manuscript does not contain toxicity assays of SpyRho on cells. Does prolonged SpyRho incubation affect cell viability?
3. The authors mention in the introduction section that the intermittent fluorescence of RhoBAST:TMR-DN is disadvantageous for STED microscopy. The imaging performance of RhoBAST:SpyRho in the STED microscopy should be compared side-by-side with RhoBAST:TMR-DN.

Reviewer #3:

Remarks to the Author:

The manuscript by Englert et al presents a new class of fluorogenic probes for aptamer-based RNA imaging that exploits spirocyclic rhodamines. The authors showed that by introducing an amide function into the probe with appropriate electron-acceptor groups the dyes show fluorogenic character on binding to the aptamer RhoBAST, developed earlier by the same group. The approach shows clear advantages compared to previously reported quencher-based fluorogenic probes in terms of brightness, fluorogenicity and cell penetration. The fast exchange rate also enabled super-resolution imaging of a target RNA in the cells using both single-molecule and STED microscopy methods. Overall, this is an excellent work, which presents a new design of fluorogenic probes for RNA aptamers, high imaging performance and compatibility with super-resolution imaging. The manuscript is clearly written and the conclusions are supported by the data. The obtained results are important and the developed probes may find applications for intracellular RNA imaging. Therefore, I recommend it for publication in Nature Communications after some revisions, as indicated below.

1) In the cellular experiments on the mRNA imaging, the 16-mer repeats of RhoBAST were used in order to achieve a good signal. The authors should clarify what is the lowest number of repeats of the aptamer needed to visualize mRNA in live cells using these new probes.

2) Single-molecule localization microscopy with exchangeable probes implies that the intermittent single-molecule emission should be observed with majority of molecules being in the off state. However, according the conditions presented in Figure 6, the concentration of the used SpyRho probe (50 nM) was only half of that used in the normal imaging conditions (e.g. Figures 2 and 3). This concentration is above K_d of the aptamer, which means that the majority of the aptamer binding sites are filled with the dye. In this case, it is not clear how on-off switching for SMLM was really achieved. It is even more puzzling that for SMLM imaging 16mer repeats of aptamer were used. This would make off-state even less probable, because at concentrations of SpyRho above K_d , most of these sites will be filled, which should make off-state per mRNA molecule theoretically impossible. The authors should clarify whether the off-state of the dyes was achieved by high irradiation power that would photobleach or turn-off majority of the dyes, as it is usually done in the STORM SMLM imaging. Moreover, the authors should provide in the supporting information the videos showing fluorescence blinking when SMLM frames were recorded. It would be also useful to provide classical information for SMLM imaging, such as number of events per second per unit area, number of photons emitted per molecule and the achieved lateral localization precision.

3) The formation of the spirolactam form in rhodamines is known to be sensitive to pH. The authors should check whether this method of RNA detection can be influenced by variation of pH close to neutral.

4) Page 5: when $K_{(on)}$ and $K_{(off)}$ values are mentioned, the reference is made to Supplementary Fig. 4, while it should be Supplementary Fig. 2.

AUTHORS' RESPONSE TO REVIEWERS' COMMENTS

Reviewer #1:

The manuscript by Englert et al reports the synthesis and characterization of a series of fluorogenic rhodamine dyes as ligands for the aptamer RhoBAST and their subsequent RNA visualization in live cells using different imaging modalities, including super-resolution microscopy. The authors systematically screened the dyes to select a bright, photostable, and fast-exchanging core, which switches on and off due to the equilibrium between the non-fluorescent spirocyclic form and the fluorescent quinoid form. The experiments are very well executed, and the manuscript is well written with excellent scholar presentation.

These probes constitute a remarkable improvement over the existing toolbox of RNA imaging dyes and will open several avenues for super-resolution RNA imaging. Notably, the RhoBAST:SpyRho system is compatible with live-cell experiments and potentially over long periods of time due to its high photostability. The authors provided good proof of the utility of their system by imaging endogenous chromosomal loci and proteins as well as labelled RNA molecules in live mammalian cells, which I haven't seen reported before. I would only suggest the authors a couple of minor revisions:

We appreciate the reviewer's overall positive evaluation of our work.

1) the introduction covers well the state-of-the-art for nucleic acid-based super-res imaging (eg DNA-PAINT), but it may be worth including a mention to peptide-PAINT strategies, including some recent advances in new fluorogenic building blocks (Angew Chem 2022, doi: 10.1002/anie.202216231);

Response 1. We thank the reviewer for this suggestion. We now mention Peptide-PAINT in the introduction and added two new references including the one mentioned.

2) I couldn't find data plots on the localization precision that is obtained in STED with this system (eg Fig 6). This would be particularly useful to know the potential application of this technology to study nanoaggregates or nanoclusters.

Response 2. We thank the reviewer for this suggestion. We believe that the reviewer meant the SMLM rather than the STED images. As suggested, we have now added localization precision analyses for **all SMLM images** shown in the manuscript including Figure 6 and Supplementary Figures 12 and 13. The mean localization precisions were found in the range 16 to 30 nm.

Overall, this is an exceptional manuscript and I recommend publication in Nature Communications. I congratulate the authors for their work. This is one of the best manuscripts I have reviewed this year.

We thank the reviewer again for his/her nice comments on our work.

Reviewer #2:

In this manuscript, Sunbul and coworkers report a novel fluorogenic probe for the RhoBAST RNA imaging system called SpyRho. This design exploits the open-closed equilibrium for fluorescence activation. Compared to RhoBAST:TMR-DN, the RhoBAST:SpyRho system features higher photostability, high brightness, higher signal-to-background ratio, and faster ligand exchange kinetics while retaining the fluorescence turn-on function upon complex formation. RhoBAST:SpyRho system is used to visualize RNA, chromosomal loci, and proteins. Super-resolution RNA imaging with SMLM and STED microscopy in live mammalian cells is also demonstrated. The study is well performed and the conclusions drawn are well supported by the data obtained in the comprehensive experiments. Super-resolution fluorescence microscopy of cellular structures provides new insights into biology. The SpyRho probe reported in the manuscript is novel and is potentially useful in super-resolution imaging. The manuscript is thus recommended for publication in the journal after the following concerns have been addressed.

We appreciate the reviewer's overall positive assessment of our work.

1. Most reaction conditions are not listed as containing an organic solvent. Is SpyRho water soluble at the high stock concentrations needed? If the stocks are dissolved in an organic solvent, what is its final concentration in the working solutions? What are the storage conditions such as temperature and concentration for SpyRho stocks?

Response 3. We have now added these pieces of information to our Methods section. In brief, we dissolve SpyRho in DMSO (up to 10 mM concentration) and store it at -20°C. In all in-vitro and cell culture experiments the final DMSO content was kept between 0.1% and 1%. SpyRho is now commercially available from the Spirochrome company and this information has also been added to the Methods section.

2. The manuscript does not contain toxicity assays of SpyRho on cells. Does prolonged SpyRho incubation affect cell viability?

Response 4. We thank the reviewer for addressing this point. Accordingly, we incubated the cells with varying concentrations of SpyRho (10 nM, 100 nM and 1 µM) for 24 h and performed a viability assay. No significant toxicity was observed. The data have been included as Supplementary Figure 7.

3. The authors mention in the introduction section that the intermittent fluorescence of RhoBAST:TMR-DN is disadvantageous for STED microscopy. The imaging performance of RhoBAST:SpyRho in the STED microscopy should be compared side-by-side with RhoBAST:TMR-DN.

Response 5. It appears that this sentence has been causing some confusion. Our intention was to clarify that the intermittent fluorescence of any aptamer:fluorophore complex with fast-exchange kinetics is disadvantageous for STED microscopy. This applies to both RhoBAST:TMR-DN and RhoBAST:SpyRho. Particularly when using low probe concentrations,

the fluorescence OFF time can be relatively long in comparison to the fluorescence ON time, which decreases the signal-to-background ratios. To circumvent this problem, high concentration of probes should be used in STED experiments. For example, in our revised manuscript, we employed 1 nM of SpyRho for SMLM experiments and 100 nM for STED experiments. Therefore, there is great demand for bright and highly fluorogenic aptamer:fluorophore pairs displaying low background fluorescence for STED microscopy. To avoid any potential misunderstanding, we have revised the sentence as follows:

“...The intermittent fluorescence of aptamer:fluorophore complexes is disadvantageous for STED microscopy, as the dark periods reduce the time-averaged emission intensity and thus the signal-to-background ratio. Higher fluorophore concentrations help shorten the dark periods but raise the fluorescence background due to the unbound probe...”

Nevertheless, we compared the performance of SpyRho and TMR-DN for imaging RNA using STED microscopy as suggested by the reviewer. We now report that STED images with SpyRho are considerably brighter than TMR-DN when taken under identical conditions, highlighting the advantage of SpyRho over TMR-DN (see Supplementary Figure 15f).

Reviewer #3:

The manuscript by Englert et al presents a new class of fluorogenic probes for aptamer-based RNA imaging that exploits spirocyclic rhodamines. The authors showed that by introducing an amide function into the probe with appropriate electron-acceptor groups the dyes show fluorogenic character on binding to the aptamer RhoBAST, developed earlier by the same group. The approach shows clear advantages compared to previously reported quencher-based fluorogenic probes in terms of brightness, fluorogenicity and cell penetration. The fast exchange rate also enabled super-resolution imaging of a target RNA in the cells using both single-molecule and STED microscopy methods. Overall, this is an excellent work, which presents a new design of fluorogenic probes for RNA aptamers, high imaging performance and compatibility with super-resolution imaging. The manuscript is clearly written and the conclusions are supported by the data. The obtained results are important and the developed probes may find applications for intracellular RNA imaging. Therefore, I recommend it for publication in Nature Communications after some revisions, as indicated below.

We appreciate the reviewer's overall positive evaluation of our work.

1) In the cellular experiments on the mRNA imaging, the 16-mer repeats of RhoBAST were used in order to achieve a good signal. The authors should clarify what is the lowest number of repeats of the aptamer needed to visualize mRNA in live cells using these new probes.

Response 6. We thank the reviewer for this suggestion. We systematically analyzed the images of live HEK293T cells expressing mAzurite-RhoBAST_n mRNA with different number of RhoBAST repeats (n = 0, 2, 4, 8 and 16). Even with only two repeats of RhoBAST, it was possible to visualize mRNA using SpyRho with a sufficiently high signal-to-background ratio (see Supplementary Fig. 9).

2) Single-molecule localization microscopy with exchangeable probes implies that the intermittent single-molecule emission should be observed with majority of molecules being in the off state. However, according to the conditions presented in Figure 6, the concentration of the used SpyRho probe (50 nM) was only half of that used in the normal imaging conditions (e.g. Figures 2 and 3). This concentration is above K_d of the aptamer, which means that the majority of the aptamer binding sites are filled with the dye. In this case, it is not clear how on-off switching for SMLM was really achieved. It is even more puzzling that for SMLM imaging 16mer repeats of aptamer were used. This would make off-state even less probable, because at concentrations of SpyRho above K_d , most of these sites will be filled, which should make off-state per mRNA molecule theoretically impossible. The authors should clarify whether the off-state of the dyes was achieved by high irradiation power that would photobleach or turn-off majority of the dyes, as it is usually done in the STORM SMLM imaging. Moreover, the authors should provide in the supporting information the videos showing fluorescence blinking when SMLM frames were recorded. It would be also useful to provide classical information for SMLM imaging, such as number of events per second per unit area, number of photons emitted per molecule and the achieved lateral localization precision.

Response 7. We greatly appreciate the reviewer for bringing this issue to our attention. We wholeheartedly agree with the reviewer's concern. When using high probe concentrations (50 nM), it can be challenging to identify the exact mechanism of intermittent single-molecule emission. It is possible that the blinking phenomenon is a result of a combination of dye-exchange, dye-switching (as seen in STORM), and dye-photobleaching caused by the high laser power. However, we would like to emphasize that we do not have precise knowledge of the intracellular free dye concentration, as the cells were only fixed and not permeabilized in our experiments.

Consequently, we repeated all SMLM experiments including circular-RhoBAST (single aptamer) and CGG-repeat containing RhoBAST (16 repeats) using only 1 nM of SpyRho. In the revised manuscript, we have now included these images (see Figure 6, Supplementary Fig. 12 and 13) as well as additional information recommended by the reviewer, such as localization precisions, the number of emitted photons per single molecule, and the number of events per second per unit area (see Supplementary Fig. 12). Additionally, we included a supplementary video that displays the fluorescence blinking events observed during acquisition of SMLM frames.

3) The formation of the spirolactam form in rhodamines is known to be sensitive to pH. The authors should check whether this method of RNA detection can be influenced by variation of pH close to neutral.

Response 8. We thank the reviewer for this suggestion. We investigated the influence of pH on the fluorescence of SpyRho and its complex with RhoBAST. We found that the fluorescence intensity of RhoBAST:SpyRho is independent of pH between 6 and 8 and slightly lower (by ~10%) at pH 5. Similarly, the fluorescence intensity of unbound SpyRho is minimally affected by pH variations from 5 to 8 (see Supplementary Fig. 5). Therefore, the RhoBAST:SpyRho system can be reliably used to visualize RNAs at different subcellular locations with pH values varying in the physiologically relevant range (pH 5 to 8).

4) Page 5: when K(on) and K(off) values are mentioned, the reference is made to Supplementary Fig. 4, while it should be Supplementary Fig. 2.

Response 9. We thank the reviewer for bringing this mistake to our attention. We have corrected the error.

Reviewers' Comments:

Reviewer #1:

Remarks to the Author:

The authors have fully addressed the minor concerns I raised in the first revision. They have also answered in detail the queries from the other two reviewers. Overall, I recommend acceptance for publication as it is.

Reviewer #2:

Remarks to the Author:

The authors have made a commendable effort in addressing the reviewers' comments and have incorporated additional experiments in the revised manuscript to enhance the persuasiveness of the data. The extra experiments have significantly improved the robustness of the results and strengthened the conclusions drawn from them. The authors have also demonstrated a thorough understanding of the issues raised by the reviewers, and have responded to them clearly and concisely. Overall, the revisions have substantially improved the quality of the manuscript, and it is recommended for publication.

Reviewer #3:

Remarks to the Author:

The authors addressed well all my concerns. Now I can recommend this manuscript for publication in the present form.